# Discovery of extended product structural space of the fungal dioxygenase AsqJ

Manuel Einsiedler [1] & Tobias A. M. Gulder [1,2] ✉

The fungal dioxygenase AsqJ catalyses the conversion of benzo[1,4]diazepine-2,5-diones into quinolone antibiotics. A second, alternative reaction pathway leads to a different biomedically important product class, the quinazolinones. Within this work, we explore the catalytic promiscuity of AsqJ by screening its activity across a broad range of functionalized substrates made accessible by solid-/liquid-phase peptide synthetic routes. These systematic investigations map the substrate tolerance of AsqJ within its two established pathways, revealing significant promiscuity, especially in the quinolone pathway. Most importantly, two further reactivities leading to new AsqJ product classes are discovered, thus significantly expanding the structural space accessible by this biosynthetic enzyme. Switching AsqJ product selectivity is achieved by subtle structural changes on the substrate, revealing a remarkable substrate-controlled product selectivity in enzyme catalysis. Our work paves the way for the biocatalytic synthesis of diverse biomedically important heterocyclic structural frameworks.

The Fe[II]/α-ketoglutarate-dependent fungal dioxygenase AsqJ from *Aspergillus nidulans* was discovered in 2014 by Watanabe et al.[1] and characterised as a key enzyme in the biosynthesis of fungal alkaloids, such as the aspoquinolones A and B (**1** and **2**, Fig. 1a)[2–4]. The non-heme enzyme catalyses a fascinating reaction cascade, involving a one-pot desaturation/epoxidation reaction of a $N^4$-methylated benzo[1,4]diazepine-2,5-dione **3** with concomitant ring-contraction/fragmentation to the viridicatin scaffold (**4**, Fig. 1b; for detailed mechanism, see Figs. S58 and S59). The single steps of this reaction cascade, as well as the function of other enzymes in the viridicatin biosynthetic gene cluster, were described in detail by different research groups[3,5–9]. In our recent study, we discovered an unprecedented additional reactivity of this enzyme[10]. While substrates with a methylene-bridged aryl substituent at $C^3$-position of the 7-membered substrate core were converted to quinolin-2(1*H*)-ones, most of the other tested substrates were transformed into quinazolin-4(3*H*)-ones, for example **5** to **6**, thereby following a new enzymatic pathway, involving α-lactam formation, hydrolysis and oxidative decarboxylation (Fig. 1c, for detailed mechanism, see Fig. S60).

Both heterocyclic product classes accessible by AsqJ are of exceptional biomedical relevance, as exemplified by the drugs levofloxacin (quinoline-4(1*H*)-one **7**) and methaqualone (quinazolinone **8**), which possess antibacterial and strongly sedative activities, respectively.

Within this study we set out to systematically evaluate the promiscuity of AsqJ, thereby not only systematically mapping its biocatalytic flexibility, but also aiming at the discovery of potential further reaction trajectories towards other biomedically relevant heterocycles. This indeed led to the discovery of further product classes addressable by AsqJ, significantly expanding AsqJ product structural space.

## Results and discussion

Given our recent findings on product selectivity of AsqJ[10], which revealed highly selective and efficient transformations of *iso*-butyl or benzyl R[1]-substituted substrates to quinazolinones and quinolones, respectively, we selected these two substitution patterns as main templates for the investigation of AsqJ substrate structure tolerance across its bimodal biocatalytic activity (Fig. 1e). These substrates were systematically altered with respect to substitution at the aromatic ring (green), configuration at the stereogenic center at $C^3$ (orange), as well

[1]Chair of Technical Biochemistry, Faculty of Chemistry and Food Chemistry, Technische Universität Dresden, Bergstraße 66, 01069 Dresden, Germany. [2]Helmholtz Institute for Pharmaceutical Research Saarland (HIPS), Department of Natural Product Biotechnology, Helmholtz Centre for Infection Research (HZI) and Department of Pharmacy at Saarland University, Campus E8.1, 66123 Saarbrücken, Germany. ✉e-mail: tobias.gulder@tu-dresden.de

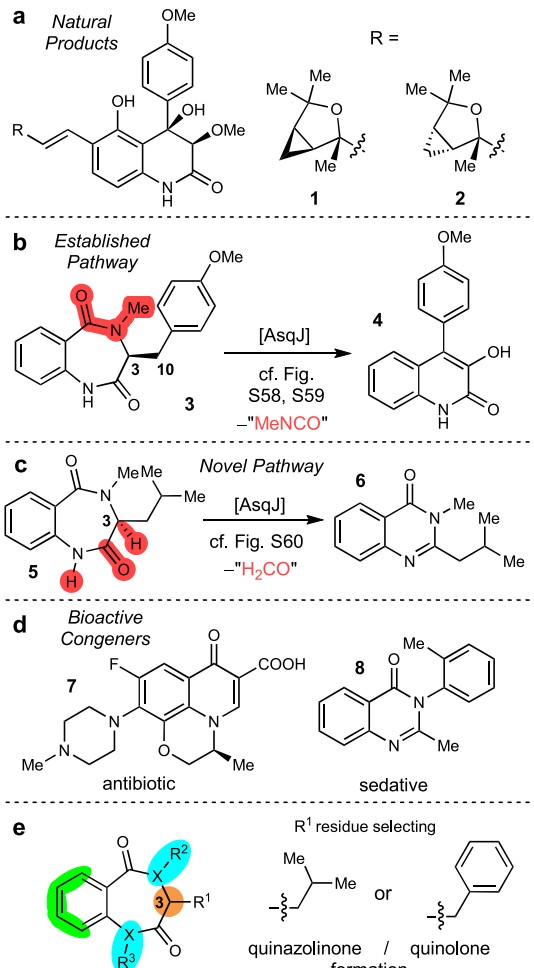

**Fig. 1 | Relevance and substrate promiscuity of AsqJ. a** Structures of the natural products aspoquinolones A (**1**) and B (**2**). **b** Biosynthetic transformation of **3** to quinolone **4** catalysed by AsqJ. **c** Pathway of AsqJ from **5** to quinazolinones **6**. The molecular portions labelled in red get excised in the course of the AsqJ catalytic transformation. **d** Examples of drugs with quinolone and quinazolinone motifs: levofloxacin (**7**) and methaqualone (**8**). **e** Graphical summary of AsqJ substrate structural alterations evaluated in this study across the quinazolinone (R[1] = *iso*-butyl) and quinolone (R[1] = benzyl) pathways.

as the nature of the heteroatom X (nitrogen or oxygen) and, for *N*-containing substrates, the substitution pattern at R[2]/R[3] (blue). In addition, tricyclic congeners in which R[1] and R[2] are connected within a five- or six-membered ring system were evaluated.

## Investigation of aromatic core substitution

The investigations were initiated by testing the promiscuity of AsqJ towards substitutions at the western aromatic core, which is unsubstituted in all natural substrates. The modifications were introduced by employing substituted anthranilic acid building blocks to enable an expedited production of diversely functionalized substrates. Since *N⁴*-methylation of the substrate was shown to be necessary for enzymatic transformation in previous studies[5,6,10], our solid phase peptide synthesis (SPPS)-based approach to benzo[1,4]diazepine-2,5-diones **9/10** was refined by loading already *Nᵅ*-methylated amino acids **11** onto Wang resin. These were easily accessible by a two-step sequence from commercially available Fmoc-protected amino acids **12**, by oxazolidinone **13** formation and subsequent reductive cleavage to **11** (see SI, chapter 2.1.1 for individual yields)[11]. After loading **11** onto Wang resin under Steglich conditions to deliver **14**, Fmoc deprotection with piperidine gave secondary amines **15**. These were coupled to

Fmoc-protected acids **16** that were synthesised from commercially available substituted anthranilic acids **17** (see SI, chapter 2.1.2 for individual yields), including fluoro- (**17a–d**), chloro- (**17e–g**), nitro- (**17h**) and methyl- (**17i–k**) substituents, as well as naphthoic acid **17l**. Couplings proceeded smoothly with HATU, HOAt and DIPEA in DMF to deliver dipeptides **18**. These were deprotected with piperidine to yield free amines **19**. Sodium *tert*-butoxide-induced cyclization with concomitant off-loading from the resin gave the desired compounds **9/10**. In case of the nitro substituted analogues **9h/10h**, these cyclization conditions led to decomposition. As the 7-membered ring can also be formed by an acid-induced cyclization[10,12], we alternatively employed 35% TFA in DCM for the synthesis of **9h/10h**, leading to the desired cyclization with concomitant cleavage from the Wang resin (Fig. 2a). Taken together, 24 substrates with modifications at their aromatic core were synthesised (Fig. 2b; see SI for further details regarding yields of intermediates etc.) and hence available for AsqJ promiscuity studies.

AsqJ substrate promiscuity was evaluated with recombinantly produced enzyme and an LC-MS-based activity assay (see SI, chapters 1.2 and 3.1 for details)[5,10]. To allow the comparison of transformations for different substrates, the assays were quenched at a maximum of 8 hours by TFA addition, even if substrate conversion was incomplete. For the pathway to quinazolinones, only fluorinated compounds **9a** to **9d** were accepted by the enzyme with clearly detectable product formation. To gain deeper insights into these conversions, we scaled up the corresponding assay with substrate **9c** and isolated all (side-)products by preparative HPLC. Beside the expected quinazolinone **21c**, putatively formed *via* **20c**, a more polar compound was also obtained (Fig. 3a, top). [1]H NMR showed that the *iso*-butyl residue in this additional product was absent, indicating that substrates entering the quinazolinone pathway can also be transformed by excision of the corresponding former amino acid side chain. This would result in formation of a quinazoline-2,4(1*H*,3*H*)-dione **22c**. While comparison of the analytical data with NMR data from the literature already strengthened this assumption[13], chemical synthesis (cf. ESI, chapter 2.3) and overlay of [1]H NMR spectra (cf. ESI, Fig. S245) unambiguously corroborated the structure of **22c**. This additional product was also detected for other substrates following the quinazolinone pathway. To further investigate the unknown mechanism of formation of this new product class, we incubated chemically synthesised **6**[10] (cf. Fig. 1c) with AsqJ to evaluate if side-chain cleavage occurs as the last step in quinazolindione formation. No conversion was observed (cf. Fig. S2), hence pointing towards a synchronous rather than a sequential formation of the quinazolindione. To corroborate this proposal, conversion of 1-[13]C-alanine-derived substrate[10] (**23**) was conducted and investigated by HRMS and NMR. In contrast to the formed quinazolinone (**24**), the corresponding quinazolindione (**25**) had the labelled carbon still present (see Fig. 3b; and SI, chapter 3.3 for details), thus clearly showing that compound class **22** is directly formed from substrate **9c** in an alternative reaction pathway (see Fig. S61 for detailed proposals on the new mechanism). This establishes quinazolindiones such as **22** as an additional heterocyclic compound class being accessible by AsqJ. The enzyme catalyses a ring contraction reaction with concomitant cleavage of the aliphatic side-chain at *C³*. This is an important finding, as the resulting product core structure is of considerable biomedical interest, as exemplified by the aldose reductase inhibitor Zenarestat (FK-366, **26**)[14] or 5-hydroxytryptamine₅A (5-HT₅A) receptor antagonist KKHT10612[15] (**27**, Fig. 3a, bottom).

For the well-established pathway towards quinolones **28**, AsqJ showed very broad promiscuity towards all substituted substrate analogues, except naphthyl-based **10l**. Especially for the fluorinated and methylated compounds **10a–d** and **10i–k**, a high up to complete conversion to desaturated (**29**) and epoxidized (**30**) intermediates was observed (Fig. 3c, exemplarily shown for **10b**). Interestingly, the final fragmentation reaction to the corresponding quinolones **28** showed

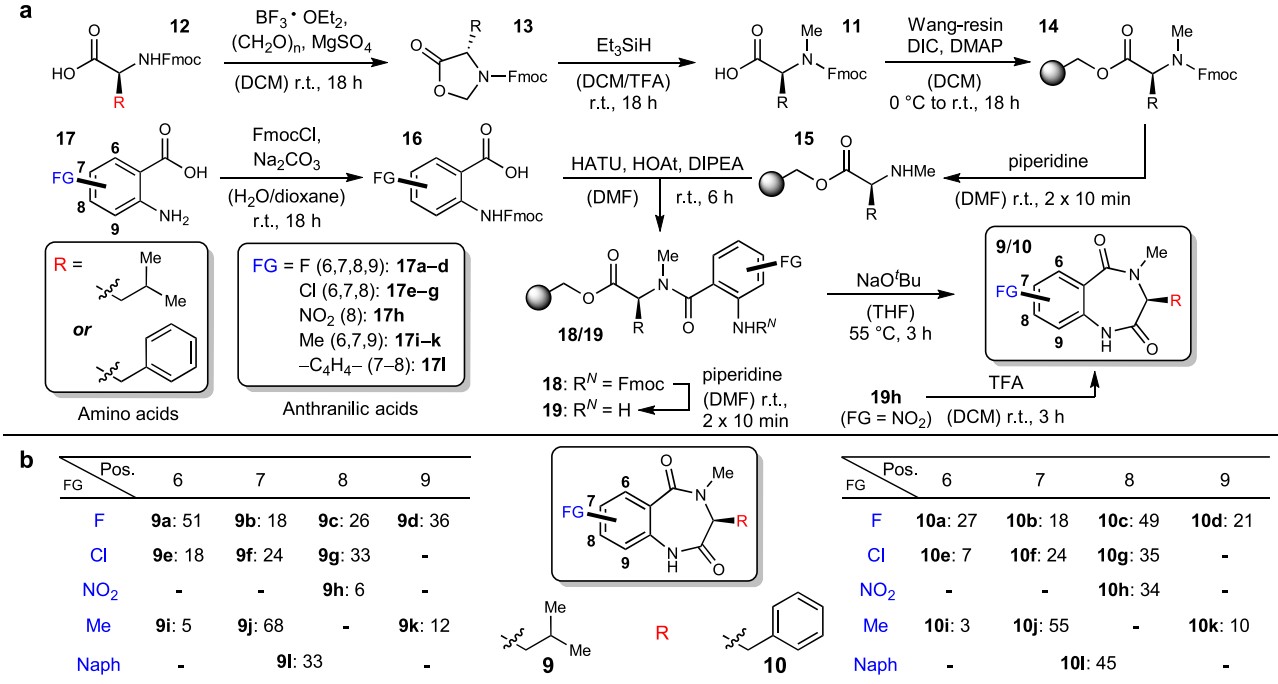

**Fig. 2 | Combinatorial substrate synthesis by SPPS. a** Synthetic route towards $N^4$-methylated benzo[1,4]diazepine-2,5-diones with different substitutions at the aromatic core, as indicated in blue. **b** Overall yields of the respective products **9/10** from **14** (over four steps). DCM dichloromethane, DIC $N,N'$-diisopropylcarbodiimide, DIPEA $N,N$-diisopropylethylamine, DMAP 4-dimethylaminopyridine, DMF $N,N$-dimethylformamide, FG functional group, Fmoc fluorenylmethyloxycarbonyl, HATU hexafluorophosphate azabenzotriazole tetramethyl uronium, HOAt 1-hydroxy-7-azabenzotriazole, TFA trifluoroacetic acid, THF tetrahydrofuran.

dependency on the aromatic substituent. For example, the assay with 7-Me-substituted substrate **10j** showed nearly complete conversion to quinolone **28j**, whereas 8-nitro compound **10h** was only converted to its epoxide **30h**. This observation can possibly be explained by the electronic effects of the corresponding substituents ($\sigma^+$ vs. $\pi^-$), hence accelerating or inhibiting the nucleophilic attack of the western aromatic ring on the strained epoxide that results in formation of the final product (cf. Fig. S59).

It is interesting to note that it appears that the rearrangement reaction from epoxides **30** to quinolones **28** is significantly accelerated by an acidic environment. To probe this, we repeated assays with **10c** and **10j** on a mg scale and monitored progress of the reaction after 90 minutes by HPLC, once after quenching by addition of TFA, once by acetonitrile (ACN) addition. The chromatograms of the assay with fluorinated compound **10c** showed the presence of desaturated (**29c**) and epoxidized (**30c**) substrate, but no formation of quinolone **28c** for both quenching methods. In contrast, the chromatogram of the assay with methylated substrate **10j** showed formation of small amounts of the final product **28j** after quenching with TFA, but not for ACN quenching (see Fig. S46). The observed difference between the substrates may result from the different substituents at the western aromatic portion, hence altered electronic effects during the fragmentation reaction (as described above). The finding that acidic reaction conditions accelerate the rearrangement from epoxide **30** to quinolone **28** goes in hand with a recent study by Chang et al., who induced the ring-contraction by addition of the Lewis acid BF$_3$•OEt$_2$[16]. It also explains the presence of a dedicated additional enzyme in the biosynthetic pathway of viridicatin scaffolds, AsqI, assisting the final rearrangement reaction under physiological conditions, as recently shown by Kishimoto et al. [2].

To further investigate the influence of reaction conditions on the final fragmentation reaction from epoxides **30** to quinolones **28**, we performed upscaled AsqJ assays with substrate **10d**. After workup by extraction with DCM, 16 different defined conditions were tested, including variations of solvent (acetone vs. chloroform vs. methanol vs.

tetrahydrofuran), reaction temperatures (22 °C vs. 50 °C) and acidic reagents (TFA vs. BF$_3$•OEt$_2$). Interestingly, most of these experiments only delivered minute amounts of the desired product **28**. For most reaction conditions, formation of a more polar side-product was observed, most prominently in CHCl$_3$ at 50 °C using TFA. HRMS analysis indicated this compound to represent 1,2-diol **31d**, originating from acid-induced activation of the epoxide followed by its nucleophilic opening by water. In methanol, the reactivity of the epoxide was strongly decreased, as little to no turnover was observed when compared to the untreated starting material. Most importantly, however, these investigations revealed that conducting the reaction in CHCl$_3$ with BF$_3$•OEt$_2$ at room temperature facilitates complete and selective conversion of **30d** to the desired quinolone **28d** (cf. Figs. S47 and S48 for details). Taken together, while TFA seems to accelerate the fragmentation reaction under aqueous (assay) conditions, BF$_3$•OEt$_2$ is the method of choice in organic solvent, with CHCl$_3$ at room temperature delivering best results in terms of both, conversion and selectivity.

Another interesting observation was made in the case of methyl substituted substrates **10i, 10j** and **10k**, where − among the respective expected products **30** and **28** − the corresponding quinazolinones **32i, j, k** were also present in the assay mixture, yet in small amounts. Quinazolinone formation apparently occurs as an alternative reaction path in these cases, since these compounds are less suitable for the natural reaction trajectory.

## Evaluation of the impact of $C^3$-configuration

In addition to all natural AsqJ substrates and all other analogues tested within this study that are derived from L-amino acids and thus are (S)-configured, we set out to investigate the enzymatic transformation with (R)-configured substrates. A previous study described that the AsqJ-catalysed reaction proceeds irrespective of the $C^3$-configuration of the substrate[8]. To corroborate these findings and furthermore test the effect of altered $C^3$-configuration on the AsqJ pathway leading to quinazolinones, the (R)-configured substrates, (R)-**9** and (R)-**10**, were also synthesised following a previously established SPPS route[10] by

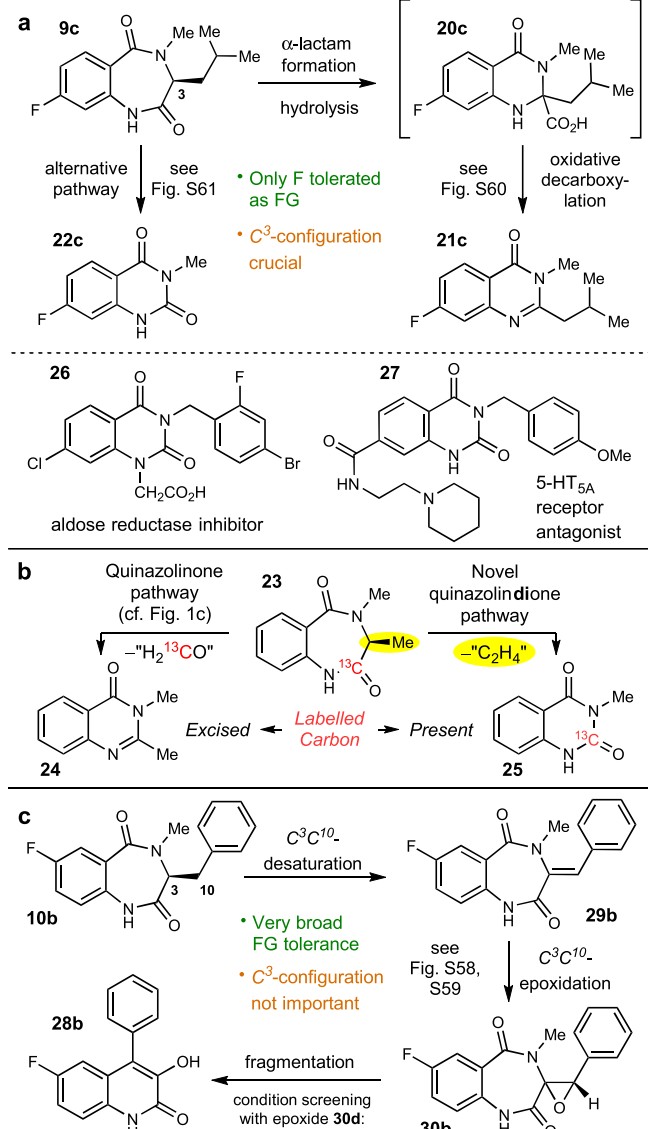

**Fig. 3 | AsqJ assay results for selected substrates. a** Summary of substrate screening outcomes for the quinazolin(di)one pathway (exemplarily shown for **9c**) and structures of quinazolindione biomedical agents **26** and **27**. **b** Difference in ring contraction for quinazolinone and quinazolindione products as observed by conversion of [13]C-labelled substrate **23**. **c** Summary of substrate screening outcomes for the quinolone pathway (exemplarily shown for **10b**). HT hydroxytryptamine.

using the respective Fmoc-protected ᴅ-amino acids (*R*)-**12** as starting materials (see SI for experimental details).

The assays with these epimerized substrates delivered interesting findings: in the case of *C*³-epimerized (*R*)-**10** complete consumption was observed in accordance with earlier investigations[8]. This goes in hand with the mechanistic proposal that the quinolone reaction pathway starts with radical formation at *C*¹⁰, followed by formation of a carbocation or a hydroxylated intermediate[7]. As a consequence, *C*³-epimerization does not affect the corresponding reaction mechanism. The assay with (*R*)-**9**, however, showed only very poor conversion. In contrast to the quinolone pathway, quinazolin(di)one formation is supposed to commence with a homolytic hydrogen abstraction at *C*³-position[10]. Epimerization of the *C*³-stereocenter hence has a direct effect on this initial radical formation and appears to strongly inhibit this reactivity (see Fig. 4). The minimal turnover still observed is rather a consequence of minute epimerization of the substrate (*R*)-**9** under assay conditions or during substrate synthesis (in particular

**Fig. 4 | Investigations of substrate stereochemical influence.** Results of enzyme assays with (*R*)-enantiomers of **9** and **10** and supposed explanation.

base-induced cyclization/off-loading reaction with NaO*ᵗ*Bu, cf. Fig. 2a) followed by transformation of the minute amounts of (*S*)-**9** thus available to AsqJ. It is interesting to note that assays with (*R*)-**10** showed formation of an additional product by HPLC analysis. While isolation was successful, unambiguous characterization based on ¹H NMR alone did not allow characterization and further NMR analysis was not possible due to very low isolated product amounts.

Taken together, it can thus be stated that *C*³-epimerization is unfavourable for both pathways, either in terms of selectivity (quinolone pathway) or efficiency (quinazolin(di)one pathway) of the enzymatic transformation.

## Exploring the effects of *N*-substitution at the central 7-membered ring

Given the requirement of AsqJ for an *N*⁴-methylated substrate for successful enzymatic conversion, investigations into its promiscuity towards other *N*⁴-alkyl functions were also targeted. Therefore, synthesis of compounds **9m** and **10m**, as well as two additional substrates **33** and **34** derived from cyclohexylalanine and (*O*Me)-tyrosine, respectively, carrying an *N*⁴-ethyl substituent was carried out, initially following the established SPPS approach (see SI for details). After the first synthetic steps, including *N*-ethylation under Mitsunobu conditions[17], which yielded the *N*ᵅ-ethylated amino acids **35**, the coupling step with Fmoc-aminobenzoic acid (**16**) failed under all tested conditions (Fig. 5). Even very potent coupling reagents, such as PyBroP or DFIH (1,3-dimethyl-2-fluoro-4,5-dihydro-1*H*-imidazolium hexafluorophosphate), did not yield the desired coupling products **36**, likely due to the increased steric hindrance caused by the *N*-ethyl group. As a workaround, the coupling was conducted with 2-nitrobenzoic acid (**37**)[18], since the nitro group exhibits less steric demand during the coupling reaction and additionally increases the acidity of the benzoic acid. Subsequent reduction of the nitro compounds **38** to the corresponding amines **39** using 2ᴍ tin(II) chloride in DMF and cyclization as described above delivered the desired *N*⁴-ethylated substrates (see Fig. 5 for individual yields)[18].

During purification of the cyclized *N*⁴-ethylated compound **10m** derived of phenylalanine, we discovered a second compound with slightly lower retention time on RP-HPLC, hence increased polarity. Isolation and HRMS/NMR analyses revealed this to be *N*¹-hydroxylated compound **40**, which results from incomplete reduction of the aromatic nitro group to hydroxylamine **41** and subsequent base-induced cyclization. This unexpected side-reaction paved the way for the additional evaluation of this *N*-hydroxylated substrate analogue **40**.

Interestingly, all *N*⁴-ethylated compounds were accepted and converted by AsqJ (Fig. 6). Enzyme reactions with **9m** and **32** proceeded following the quinazolinone pathway to *N*³-ethylated

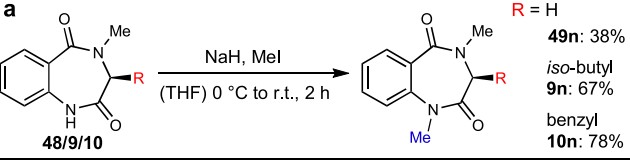

**Fig. 5 | Optimized synthesis of *N⁴*-ethylated substrates[18].** Yields are given over seven steps, starting from loaded resin. DFIH 1,3-dimethyl-2-fluoro-4,5-dihydro-1*H*-imidazolium hexafluorophosphate, PyBroP bromotripyrrolidinophosphonium hexafluorophosphate.

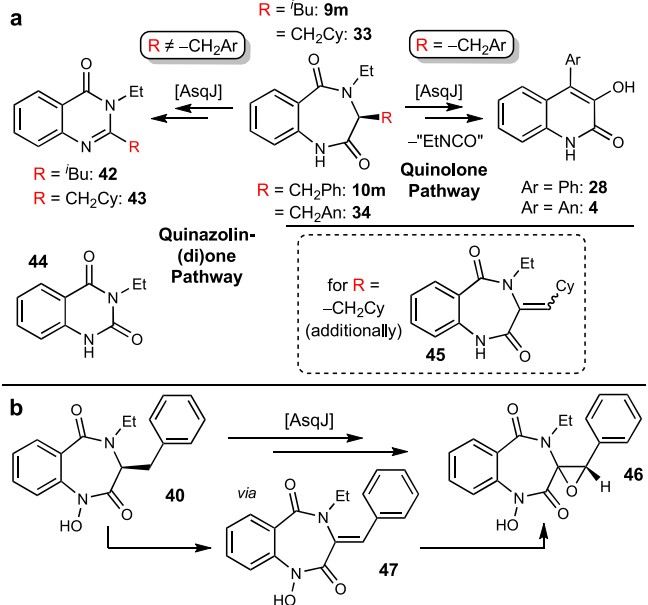

**Fig. 6 | AsqJ assay results with *N⁴*-ethyl analogues. a** Biocatalytic transformation of *N⁴*-ethylated substrates. **b** Conversion of the additionally *N¹*-hydroxylated analogue **40**, where no quinolone formation was observed. Cy cyclohexyl, An 4-anisolyl (*p*-methoxyphenyl).

quinazolinones **42** and **43** (with observation of presumed corresponding quinazolin-2,4-dione **44**, and desaturated side-product **45**), while **10m** and **34** were converted following the quinolone mechanism with concomitant elimination of ethyl isocyanate. The *N¹*-hydroxylated

**Fig. 7 | Synthesis and conversion of *N,N*-dimethylated substrates. a** Late-stage methylation (blue) towards the desired compounds. **b** Results of enzymatic conversion with mechanism proposal for hydroxylation of **49n** (proposed origin of hydroxyl group labelled in green), as well as structure of blockbuster pharmaceutical lorazepam (**51**).

compound **40** was accepted as a substrate, but conversion stopped at the stage of the epoxide **46** via desaturated compound **47**, comparable to the nitro compound **10h**. This observation fits to the mechanistic proposal that an abstractable hydrogen at the *N¹* atom facilitates the fragmentation reaction[5].

In addition to *N⁴*-alkylation, the effect of *N¹*-methylation on AsqJ catalysis was explored. Optimization of a test reaction with commercially available glycine-derived benzo[1,4]diazepine-2,5-dione **48** yielded the dimethylated compound **49n** in 38% yield by deprotonation with sodium hydride followed by treatment with methyl iodide in THF. Following this approach, compounds **9** and **10** were converted to their double methylated congeners **9n** and **10n** in 67% and 78%, respectively (Fig. 7a).

Enzymatic conversion of the *N¹*-methylated substrate **49n** (in which R = H) led to formation of a single product (Fig. 7b). Upscaling and isolation allowed investigation of its structure by ¹H NMR and HRMS, which showed that the *C³*-hydroxylated compound **50** was formed. This defines yet another reactivity accessible by AsqJ catalysis, leading to a product that has structural similarity to the blockbuster pharmaceutical lorazepam (**51**). We assume that the *N¹*-methyl group blocks the *N¹–C³* bond formation of the quinazolinone pathway, hence leading to an intermediate free carbon-centred radical or carbocation **I1** (after a second radical abstraction, cf. Fig. S61) at *C³*, which is ultimately trapped by water (free or enzyme-bound). To mechanistically investigate whether this is an enzymatic or non-enzymatic process, stereochemical analysis of **50** was of high interest. While an enantio-pure or -enriched product would hint towards an enzymatic hydroxylation, a racemic mixture would rather point towards non-enzymatic capture of an intermediate radical or ion (**I1**) by a water molecule. It has to be taken into account, however, that 3-hydroxylated diazepines such as **50** are highly prone to epimerization. For example, enantiopure lorazepam (**51**) was shown to have a half-live time of about 21 minutes[19]. We therefore investigated the optical purity of freshly prepared **50** by rapid analysis by HPLC on a chiral phase. These investigations revealed **50** to occur as racemic mixture, pointing at a non-enzymatic quenching reaction or very fast loss of stereochemical integrity due to racemization (cf. ESI, chapter 3.3; Fig. S44). To further distinguish between these possibilities, we performed an enzyme assay with substrate **49n** and H₂¹⁸O as reaction solvent (approx. 80 vol-%). By HRMS, we observed that approx. 36% of **50** were indeed ¹⁸O-labelled, as indicated by the corresponding mass counts (cf. ESI, Fig. S45). These

results further substantiate the hypothesis that the hydroxyl group originates from water quenching of an enzyme-activated intermediate, such as **I1** (Fig. 7b).

For the $N^1$-methylated substrate **9n**, the assay showed only trace consumption of substrate with formation of unidentified products in minute amounts. In the case of **10n**, epoxidized compound **30n** was observed, as well as a small amount of fragmentation product **28n**. Moreover, an additional, large product peak was detected by HPLC analysis. Up-scaling of the reaction and isolation of this compound was attempted to facilitate downstream NMR and HRMS analyses. However, its characterization was not possible due to product instability. Taken together, it can be stated that $N^1$-substitution is not favourable for either the quinazolinone or the quinolone pathway, yet paves the way for a new biocatalytic aliphatic *C*-hydroxylation in glycine-derived substrate analogue **49n**.

## AsqJ biocatalytic fidelity towards tricyclic substrates

As a further expansion of the possible substrate scope, we produced compounds in which the R[1] and R[2] residues are part of a five- or six-membered cyclic ring system, synthetically derived from proline (**52**) or pipecolic acid (**53**), respectively. The synthesis was accomplished in 44% (**54**) and 83% (**55**) following the solid-phase peptide synthesis (SPPS) approach described earlier (Fig. 8a)[10]. This resulted in a very fast synthetic access to this compound class by loading of the Wang resin to give esters **56/57**, Fmoc deprotection to secondary amines **58/59**, coupling with **16** to the dipeptides **60/61**, second deprotection to aromatic amines **62/63** and base-induced cyclization to **54/55**. To our delight, although with overall lower turnover (cf. quantification of the transformation of key substrates below), enzymatic conversion of tricyclic substrate **54** proceeded to give tricyclic quinazolinone natural product deoxyvasicinone (**64**, Fig. 8b), a compound reported to be found in, e.g., *Peganum harmala* L. seeds[20]. This result emphasizes the versatility of AsqJ, underlining its potential use in chemo-enzymatic natural product synthesis. Interestingly, compound **64** represents the carbon scaffold of the plant-derived natural products vasicinone (**65**) from *Adhatoda vasica* Nees[21] and its regioisomer isovasicinone (**66**). **65** and **66** show antifungal activity against, e.g., *Fusarium graminearum* and *Helminthosporium sativum*[22]. To further proof the applicability of AsqJ in bioactive natural product synthesis, we aimed at the chemo-enzymatic synthesis of isovasicinone (**66**). Starting with hydroxyproline **67**, Fmoc- and TBS-protection at the amine and hydroxy functionalities delivered building block **68** (cf. ESI 2.1.2), which was incorporated into hydroxylated precursor **69** (cf. ESI 2.2.1) (Fig. 8d, top).

AsqJ assays with this substrate indeed led to formation of the desired antifungal compound isovasicinone (**66**). In addition, a small amount of side-product was observed. Isolation and NMR characterization was not possible because of the very small amounts produced, but HRMS indicated a loss of hydrogen and water when compared to **69**. We therefore propose this compound to be pyrrole **70**, whose formation can be explained by initial AsqJ-catalysed dehydration of **69** to **71** followed by elimination of the secondary alcohol to deliver the aromatic system (Fig. 8d).

Application of L-pipecolic acid-derived substrate **55** led to formation of quinazolinone **72**, representing the natural product mackinazolinone isolated from *Mackinlaya* sp[23]. In addition to **72**, formation of desaturated side-product **73** was observed. Moreover, both assays showed formation of an additional, more polar side-product. Unambiguous structure elucidation after attempted isolation of this side-product in the conversion of **54** was not possible due to the minute amounts formed, but [1]H-NMR and HRMS indicated this substance to be the intermediate acid (**74** and analogously **75**) of the quinazolinone pathway. As potential alternatives to formation of side-product structures **74/75**, the above described, newly discovered side-chain excision mechanisms for bicyclic substrates leading to

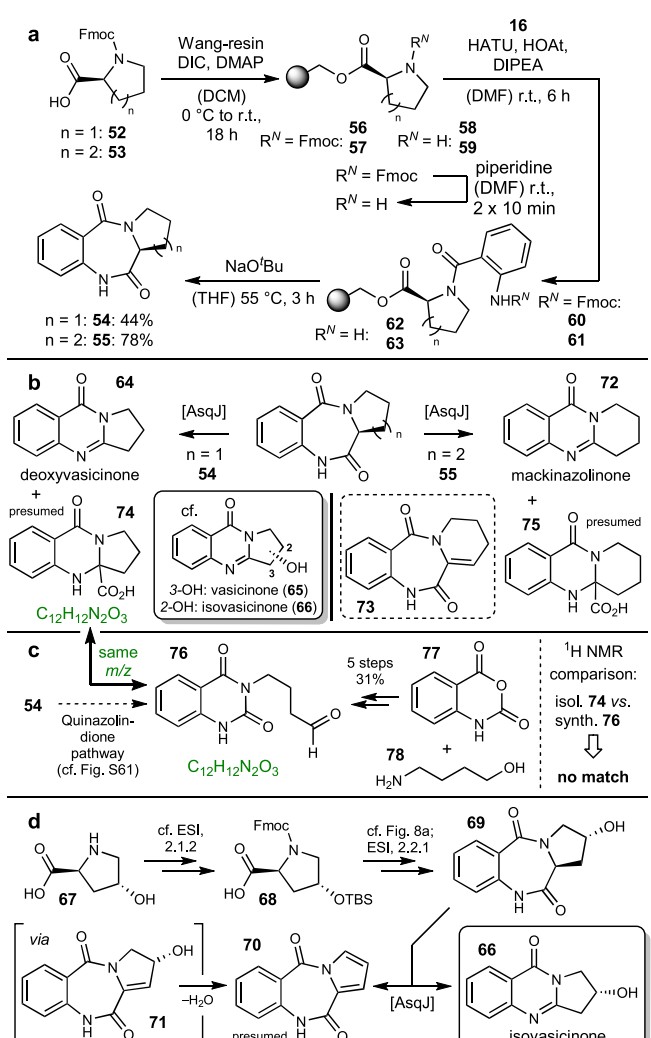

**Fig. 8 | Investigations involving tricyclic substrates. a** Synthesis of tricyclic substrates **54/55. b** Enzymatic conversion. **c** Synthesis of hypothesized aldehyde **76** and comparison with isolated compound **74**. Green colour indicates same chemical formulas and hence *m/z* ratios of these compounds. **d** Synthesis and conversion of L-4-hydroxyproline-derived substrate **69**. NMR nuclear magnetic resonance, TBS *tert*-butyldimethylsilyl.

quinazolindiones, e.g., **22c** (cf. Fig. 3a/b) were also considered. With tricyclic substrates, this reaction would potentially lead to ring opening of the five- (**54**) or six- (**55**) membered eastern ring originating from the cyclic amino acids **52/53**. Following our mechanistic proposal for the quinazolindione pathway (cf. Fig. S61), in the case of **54** this would furnish the bicyclic aldehyde **76** (cf. Fig. 8c). As the molecular masses of the proposed acid intermediate **74** compared to the resulting, hypothetical **76** would be identical, we desired to experimentally test this alternative hypothesis. Therefore, we conducted the synthesis of aldehyde **76** in a five-step route starting from isatoic anhydride (**77**) and 4-aminobutanol (**78**) in an overall yield of 31% (cf. ESI, chapter 2.3 for experimental details). Comparison of the spectroscopic data ([1]H NMR) of the observed side-product compared to synthetic aldehyde **76** showed no match (cf. Fig. S246), hence further substantiating this compound to indeed correspond to acid **74**.

## Probing the exchange of the nitrogen (amide) by oxygen (ester) heteroatoms

Moreover, replacement of the amide moieties in **9/10** as esters to generate oxygen-containing heterocyclic compounds by enzymatic

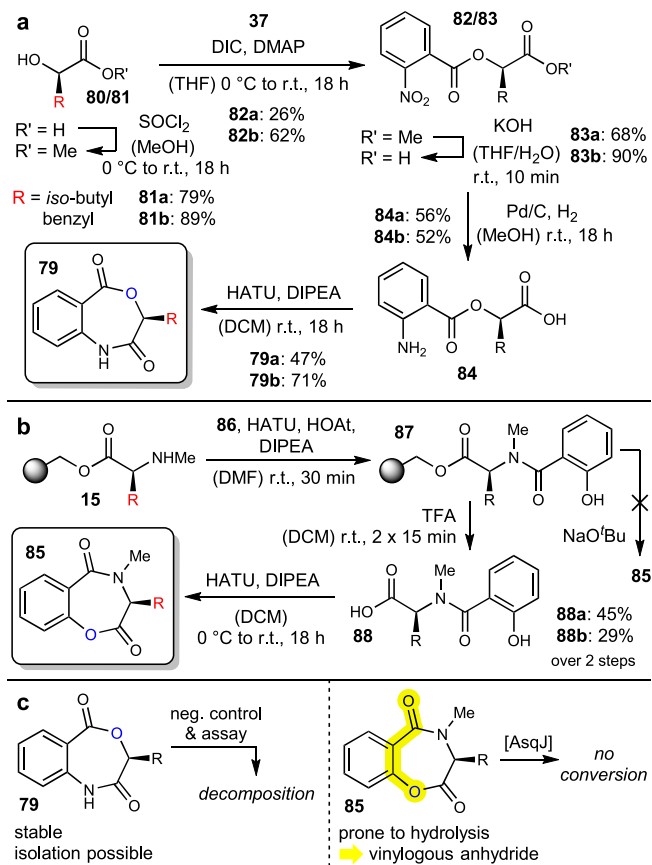

**Fig. 9 | Syntheses and assays of ester substrate analogues. a** Route to benzo[*e*]oxazepines **79**. **b** Synthesis of labile benzo[*f*]oxazepines **85**. **c** Observations during attempted enzymatic conversion.

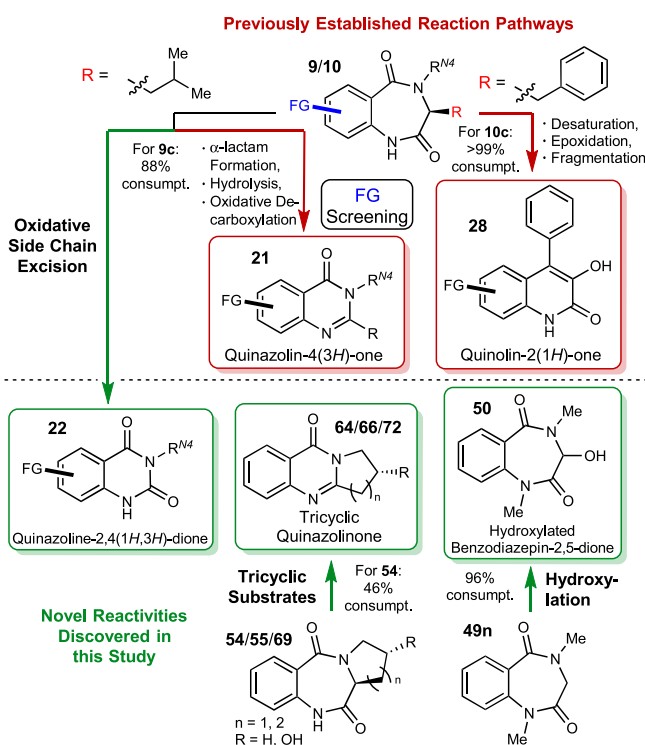

**Fig. 10 | Substrate and product map of AsqJ.** Overview over compounds/compound classes accessible by enzymatic conversion with AsqJ, as extended (red) and discovered (green) within this study. (Proposed) single reaction types are given with the reaction arrows.

conversion using AsqJ deemed interesting to us. For the northern substitution (towards **79**), we developed a short reaction sequence starting from lactic acids **80** (Fig. 9a). After methyl ester protection with thionyl chloride to **81**, esterification with 2-nitrobenzoic acid (**37**) under Steglich conditions delivered esters **82**. After saponification with KOH to the free acids **83** with short reaction time to avoid cleavage of the desired ester, and Pd-assisted hydrogenation of the nitro group to amino acids **84**, HATU-mediated cyclization afforded cyclic amides **79** (see Fig. 9a for individual yields). For the southern esters (**85**), resin-bound N-methylated amino acids (**15**) were coupled to salicylic acid (**86**) under standard conditions (HATU, HOAt, DIPEA) in a reduced amount of time to avoid undesired esterifications, affording phenols **87**. Unfortunately, base-induced cyclization of **87** was unsuccessful with the alcohol instead of the amine. Cleavage from the resin with 35% TFA in DCM to the corresponding hydroxy acids **88** before HATU-assisted cyclization delivered the desired products **85** (see Fig. 9b for individual yields). However, these esters were unstable during purification, presumably because of the conjugated acid motif, imparting reactivity of vinylogous anhydrides (cf. Fig. 9c, labelled in yellow). Therefore, these substances were subjected to AsqJ directly after workup without further purification.

Unfortunately, ester surrogates **79** proved unstable under the assay conditions, with even the enzyme-free negative control showing decomposition (Fig. 9c). On the other hand, compounds **85** (freshly prepared) were observed to be reasonably stable in this environment, but no difference between negative control and AsqJ assay was observed. This may be a result of impurities contained in the crude substrate mixture that had to be employed due to the instability of **85** (see above), which might have inhibitory effects on

AsqJ, or, more likely, be a consequence of the altered electronic and geometric properties of the ester *vs.* the natural amide substrates. Overall, these experiments lead to the conclusion that the central benzo[1,4]diazepine-2,5-dione motif is a required general precondition for AsqJ substrates.

## Quantification of the transformation of key substrates

To investigate and compare AsqJ substrate preferences of the established and the newly discovered reactivities, we performed HPLC-based monitoring of substrate consumptions. We selected one representative substrate with high conversion for each of the described reaction trajectories towards different product classes. We hence utilized substrates **9c** (quinazolin(di)one pathway), **10c** (quinolone pathway), **49n** (hydroxylation reactivity) and **54** (tricyclic substrate following quinazolinone pathway) for tracking of time-dependent substrate consumption by AsqJ. For each of these substrates, HPLC-UV concentration calibration curves at 220 nm were recorded and used to analyse substrate consumption in individual assays after 15, 60, 180 and 360 minutes (cf. ESI, chapter 3.4). This revealed that natural phenylalanine-derived substrate **10c** was converted very fast, with 50% consumption after less than 40 minutes and more than 95% after 180 minutes. Hydroxylation of dimethylated glycine-derivative **49n** and quinazolin(di)one formation from leucine-derived **9c** were slower (50% consumption after approx. 100 and 120 minutes, respectively). However, after 360 minutes, conversion of approx. 90% was detected for both substrates. For proline-derived tricyclic **54**, conversion was generally slower and overall substrate consumption only reached 46% after 360 minutes. Overall, these investigations unsurprisingly revealed that AsqJ is most competent for transformation of its original, natural precursor structures such as **10c**, but that catalytic efficiency also remains very high (for **9c, 49n**) to sufficient (**54**) for its biocatalytic application across all four investigated transformations (Fig. 10).

Within this work, we systematically evaluated the substrate promiscuity and product spectrum of the fungal dioxygenase AsqJ. While the experiments confirmed mechanistic proposals for the pathways representing the established bimodal activity of AsqJ, only quinolone formation turned out to be highly flexible, while quinazolinone formation was proven to have a rather narrow scope concerning the tested structural alterations of the substrate. The central seven-membered diamide structure was shown to be a necessary precondition for both, AsqJ enzymatic conversion and substrate stability. Moreover, by our broad-spectrum (cf. Table S1) substrate testing, we herein discovered the capability of AsqJ to generate tricyclic quinazolinones from tricyclic benzo[1,4]diazepine-2,5-diones, which can be applied in the synthesis of natural products, such as vasicinone (65). This is a highly interesting example of an enzyme that can be utilized to build a bridge between the biosynthesis of secondary metabolites of different kingdoms, as directly exemplified by the AsqJ-catalysed formation of deoxyvasicinone (64), isovasicinone (66) and mackinazolinone (72).

Most importantly, our in-depth studies moreover revealed another heterocyclic compound class accessible by AsqJ, namely quinazolin-2,5-diones 22/44. Mechanistic studies involving a $^{13}$C-isotope labelled substrate 23 revealed this product class to be formed simultaneous to the corresponding quinazolinones and led us to the assumption that the side chain is oxidatively excised from the starting material (see Fig. S61 for a mechanistic proposal). Additionally, the biocatalytic formation of hydroxylated compounds such as 50 from unsubstituted (with respect to R$^1$, Fig. 1e) substrates opens the door for further studies towards the application of AsqJ for the synthesis of bioactive diazepines (as indicated by the structural similarity of 50 to lorazepam (51)). Taken together, the extensive substrate promiscuity and resulting product diversity (cf. Table S2), as uncovered within this study, highlight a remarkable substrate-controlled product selectivity switch in the fascinating dioxygenase AsqJ. This work thus establishes AsqJ as a highly variable and sustainable tool for the generation of a whole set of bioactive compounds with diverse heterocyclic core structures by only subtle structural changes of the employed substrates (Fig. 10).

Overall, our work has comprehensively mapped the substrate scope across the two previously known AsqJ reaction pathways and has resulted in the discovery of the biocatalytic competence of AsqJ to transform tricyclic substrates in addition to establishing two additional reactivities: side-chain excision to quinazolin-2,5-diones and substrate $C^3$-hydroxylation. Further work towards the optimization of reaction conditions to enable large-scale preparative application of AsqJ is currently ongoing in our laboratories.

## Methods
### General procedure of analytical enzyme assay
Analytical enzymatic assays were carried out in a total volume of 120–350 µL, containing 1 mM of the substrate (dissolved in DMF: 100 mM), 50 µm (5 mol-%) of purified AsqJ, 2.5 mM α-ketoglutarate, 4 mM ascorbic acid, 100 µM iron(II) sulphate and 5% (v/v) DMF. The reaction buffer contained 50 mm TRIS HCl at a pH of 7.4. After incubation for 5–6 h (8 h max.) with shaking at 300 rpm, the enzyme was precipitated by addition of TFA (1 µL TFA/40 µL assay) and removed by centrifugation for 5 min at 9700 × g. The supernatant (20 µL) was analysed by HPLC (Knauer Azura) at 220 nm with coupled ESI-MS (Advion, single-quadrupole mass analyser).

### Experimental details and data
Experimental procedures, like enzyme expression and purification, synthetic details, enzyme transformations for structure elucidation, as well as analytical characterization of compounds, comprising chromatograms, UV and NMR spectra are provided in the supplementary information.

## Data availability
The authors declare that the data supporting the findings of this study are available within the paper and its supplementary information file, or from the corresponding author upon request.

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

## Acknowledgements
This project was generously funded by the German Research Foundation (DFG, GU 1233/1-1 to T.A.M.G.). The authors thank Dr. T. Lübken (TU Dresden, Organic Chemistry I) for measuring NMR spectra and advice with HPLC measurements. The help of M. Wiegand (TU Dresden, Chair of Technical Biochemistry) with large-scale production of AsqJ is thankfully acknowledged. Open Access funding enabled and organised by Projekt DEAL.

## Author contributions
M.E. conducted all experimental work. T.A.M.G. supervised the entire project. Both authors designed the experimental work, analysed the data, discussed the results, and wrote the manuscript.

## Funding

## Competing interests
The authors declare no competing interests.
