## [Peer review file · Nature Communications]

REVIEWER COMMENTS

Reviewer #1 (Remarks to the Author):

In the study "Discovery of extended product structural space of the fungal dioxygenase AsqJ", Einsiedler and Gulder have performed a broad activity screening of the dioxygenase AsqJ against several functionalized substrates produced via solid-/liquid-phase peptide synthesis.

The authors find that that AsqJ catalyse in addition to the established quinolone and quinazolinones pathways, also the formation of quinazolidiones, important compounds of biomedical relevance. The findings support the general notion in the field (partly put forward by the authors in previous work), that AsqJ shows significant catalytic promiscuity.

The authors characterize the products via LC-MS, HPLC and NMR, and perform an impressive quantification of the various pathways. The methods are well suited for screening. However, I find that the study lacks detailed mechanistic experiments of the various reaction pathways, and it does not explain how the properties of catalytic site or surroundings may favour one of the reactions.

Overall, I find that the study is well-performed and interesting, particularly with regard to the new reaction pathways, but too specialized and difficult to follow for the broad audience of Nature Communications. I therefore suggest that the authors submit the manuscript to a more specialized chemical journal.

Reviewer #2 (Remarks to the Author):

Kenji Watanabe, Ph.D.

Department of Pharmaceutical Sciences

University of Shizuoka

City of Shizuoka, 422-8526, Japan

Telephone: +81-54-246-5662

Fax: +81-54-246-5666

email: kenji55@u-shizuoka-ken.ac.jp

December 13, 2022

Dr. Majda Bratovic

Associate Editor in Nature Publications

Nature communications

Dear Dr. Majda Bratovic,

The manuscript (Manuscript ID: NCOMMS-22-50046) by Tobias A. M. Gulder, et al. entitled “Discovery of extended product structural space of the fungal dioxygenase AsqJ” explores the expanded substrate promiscuity of the fungal dioxygenase AsqJ towards a broad variety of substrates. Efficient synthesis routes based on solid and liquid phase peptide synthesis (SPPS/LPPS) or combinations thereof are developed. These routes facilitate the rapid synthesis of a broad range of functionalized substrates, for example with different substituents at the aromatic core structure, C3-epimerization, N-alkylation pattern, exchange of heteroatoms (nitrogens replaced by oxygens), as well as tricyclic substrates. This in turn enables for the first time an in-depth screening of the established natural quinolone pathway, where a broad tolerance towards different substituents at the core aromatic portion is shown, enabling the use of AsqJ for the efficient biocatalytic synthesis of quinolones and their epoxide precursors in future applications. Tricyclic precursors are transformed into the corresponding tricyclic quinazolinones following the recently described unnatural reactivity of AsqJ. This finding is a fascinating example of a fungal enzyme that produces a compound closely related to a plant natural product, namely vasicinone, thus from a different kingdom of life. It is even more captivating that this outcome is triggered by the substrate structure. Most importantly, the presented investigations reveal that AsqJ additionally gives access to yet unknown heterocyclic product classes of high biomedical importance. These are quinazolindiones (formed by oxidative cleavage of the aliphatic side chain of the substrate following a new mechanism described here for the first time) as well as hydroxylated benzodiazepindiones, which show high similarity to the blockbuster pharmaceutical lorazepam.

Taken together, the paper contains a large set of well-planned and executed experiments that reveal valuable biocatalytic applications of AsqJ and lead to the discovery of two novel reaction pathways. All claims are well substantiated by the reported analytical data. The discoveries add substantial knowledge on the substrate promiscuity of AsqJ and its substrate-directed product outcome, which lays the foundation for the development of biocatalytic synthetic access to a broad range of biomedically

important heterocyclic small molecules using AsqJ. Therefore, this work will be interesting for a broad range of the versatile readership of Nature Communications.

The following minor points could be addressed to make the paper even more accessible:

1. To facilitate a quick and easy digest of the large number of biocatalytic results presented in the manuscript, it would be advantageous to generate a table that graphically captures the product outcome for each substrate in a single picture. This could be added to the ESI.
2. Similarly, such a tabulated overview on overall yield of the synthesized substrates would be a nice addition to the ESI.
3. One of the new reactions described in the paper leads to C-hydroxylation of substrate 48n, thereby generating a stereogenic center at C3 in the product. It would be interesting to know if this is a stereoselective process. This reaction could be scaled up and the formed product thoroughly analyzed, for example by product separation using HPLC on chiral material. The results of these investigations would also help further substantiate the mechanistic proposal via radical/ionic intermediates.
4. For the transformation of the proline-derived substrate 53, it is hypothesized the more polar side-product (in addition to tricyclic quinazolinone 63) to likely be the carboxylic acid 67. Regarding the side chain excision that happens for other substrates leading to quinazolinindiones, this reaction would lead to ring opening of the five-membered proline ring, setting free a bicyclic aldehyde. As the molecular masses of the proposed 67 compared to such an alternative, hypothetical aldehyde product would be identical, it would be interesting to exclude such a reaction pathway. This could for example be achieved by synthesis of the bicyclic aldehyde and comparison of the spectroscopic data of the isolated intermediate versus the synthesized compound. These investigations would either corroborate the mechanistic proposal for quinazolinindione formation, or further substantiate the hypothesis currently stated in the manuscript.

Sincerely,

Kenji Watanabe

Reviewer #3 (Remarks to the Author):

The manuscript authored by Einsiedler and Gulder used a series of substrate analogs to explore the substrate scope and reactivity of AsqJ. Following substrate preparation and enzymatic assays, HPLC, product isolation and NMR were carried out to characterize reaction products. Through changing the substituent from phenyl, isobutyl, methyl to proton (along with N-substitution), AsqJ showed diverse reactivity. Total four outcomes were described in which two of them were described in the previous studies. Two new reactivities are associated with quinazolindione and hydroxylation. The authors claimed that some of these new reactions are important since several of them are affiliated with currently used drugs. This reviewer appreciates the authors' efforts in synthesizing substrate analogs and exploiting the substrate promiscuity toward AsqJ. On the other hand, this reviewer has some concerns and questions regarding to what are new knowledge and insight that we readers can learn from this manuscript.

1. To emphasize the usefulness of this enzymatic reaction toward biomedical applications, the author should consider demonstrate how to apply this reaction in making any of the listed compounds (at least one or two examples), e.g. 26, 27, 50 and 64.

2. Possible mechanisms regarding to quinazolindione was proposed (Figure S45), while hydroxylation was briefly discussed. Because of the majority of enzymes in this super family are classified as hydroxylases. Is hydroxylation associated with this reactivity or is it a result of quenching of cation? While a brief discussion was provided, the author should consider designing experiments to distinguish these possibilities. Notably, the chirality of the hydroxylation seems not determined. The authors should also consider determine the stereochemistry.

3. The authors should provide kinetic parameters of these reactions. Based upon the HPLC traces the current description, it is difficult for readers to capture the reaction preference. At least, TON (the turnover number) and TOF (the turnover frequency) of some key substrates should be provided.

4. The authors explained the observed rearrangement trend, e.g. 30 to 28, using electronic effect. This is an interesting observation, but should be investigated with caution. The authors also mentioned that this rearrangement can be accelerated under different conditions, e.g. BF₃. To better correlate the observed outcome, it might be helpful to try rearrangement under the reported condition using the available epoxides, e.g. 30. In addition to acidity, other factors, e.g. solvent, temperature and etc, might influence this reaction.

Tabular overview on changes made based on the valuable comments of the reviewers

Reviewer #	Comment #	Reviewer Comment
1	1	I find that the study lacks detailed mechanistic experiments of the various reaction pathways, and it does not explain how the properties of catalytic site or surroundings may favour one of the reactions.
Reply: We fully agree with the reviewer that mechanistic insights into the different mechanisms are important. We therefore provided detailed mechanisms for the well-investigated natural AsqJ pathway to quinolones (see ESI, Figures S57 and S58) as well as for the new pathway to quinazolinones (see ESI, Figure S59) in addition to mechanistic proposals for substrate hydroxylation (see Figure 7b, main manuscript) and side-chain cleavage to quinazolindiones (see ESI, Figure S60). Furthermore, we have experimentally evaluated the impact of C3 stereochemistry on quinolone versus quinazolin(di)one pathways (see main manuscript, Figure 4 and respective text, page 4, lines 16ff.). Within the revised work, we also provide additional investigations on the stereochemical outcome of the hydroxylation reaction from substrate 49n to 50 (see main manuscript page 5; ESI, chapter 3.3; cf. replies to comments 2.3 and 3.2). In addition, we provide experimental mechanistic insights into quinazolindione formation, thereby validating that this indeed is a separate pathway and not a result of further modification of initially formed quinazolinones (tested by assays with quinazolinone 6, see main manuscript, page 2, lines 61 ff.) and determined which molecular portions get excised exactly by conducting experiments with ¹³C-labelled substrate 23 (see Figure 3b and text main manuscript, page 3, line 1), which overall resulted in our mechanistic proposals for quinazolindione formation (see ESI, Figure S60). We thus believe that sufficient insights into mechanistic questions are now provided for this manuscript. In depth analyses of the impact of catalytic residues in the enzyme active site or its proximity are not part of the current manuscript that already contains a tremendous amount of experimental work and insights. This analysis by itself will be very extensive and will thus be evaluated in future work in our laboratory, which will build on the current manuscript and will aim at changing AsqJ selectivities by strategic active-site mutations.		
2	1	To facilitate a quick and easy digest of the large number of biocatalytic results presented in the manuscript, it would be advantageous to generate a table that graphically captures the product outcome for each substrate in a single picture. This could be added to the ESI.
Reply: We thank the reviewer for this excellent idea. We added the corresponding table to the ESI at the end of chapter 3.2 (Table S2).		
2	2	Similarly, such a tabulated overview on overall yield of the synthesized substrates would be a nice addition to the ESI.
Reply: Again, thank you very much for this suggestion that will make our results even more accessible. We added the corresponding table to the ESI at the end of chapter 2.2 (Table S1).		
2	3	One of the new reactions described in the paper leads to C-hydroxylation of substrate 49n , thereby generating a stereogenic center at C3 in the product. It would be interesting to know if this is a stereoselective process. This reaction could be scaled up and the formed product thoroughly analyzed, for example by product separation using HPLC on chiral material. The results of these investigations would also help further substantiate the mechanistic proposal via radical/ionic intermediates.
Reply: We thank the reviewer for this excellent suggestion. We now added these exact experiments as suggested and investigated the corresponding AsqJ product 50 (from 49n) by HPLC on a chiral phase (see ESI, chapter 3.3, Figure S44; manuscript, page 5, lines 43 ff.). With this analysis, we observed a ~1:1-mixture of enantiomers of 50, which might hint at a non-enzymatic quenching mechanism. However, it also has to be taken into account that 3-hydroxylated diazepines, such as lorazepam (51), appear to have a very short epimerization half-life time, which was shown to be about 21 minutes for 51.¹ While we used freshly prepared 50 and conducted downstream chiral analysis very quickly, it cannot be excluded that stereochemical information of an initially stereoselective process is lost before being determinable.		

2

4

For the transformation of the proline-derived substrate **54**, it is hypothesized the more polar side-product (in addition to tricyclic quinazolinone **64**) to likely be the carboxylic acid **74**. Regarding the side chain excision that happens for other substrates leading to quinazolindiones, this reaction would lead to ring opening of the five-membered proline ring, setting free a bicyclic aldehyde. As the molecular masses of the proposed **74** compared to such an alternative, hypothetical aldehyde product would be identical, it would be interesting to exclude such a reaction pathway. This could for example be achieved by synthesis of the bicyclic aldehyde and comparison of the spectroscopic data of the isolated intermediate versus the synthesized compound. These investigations would either corroborate the mechanistic proposal for quinazolindione formation, or further substantiate the hypothesis currently stated in the manuscript.

Reply: This is an important observation by the reviewer and we were happy to address these questions, thereby testing our hypothesis. We now performed the synthesis of the suggested alternative aldehyde side-product **76** (see ESI, chapter 2.3.). Comparison of the NMR spectrum of **76** with that of the isolated compound of the assay with proline-derived substrate **54** (see ESI, Figure S245) shows no match between these two compounds. This corroborates our initial proposal that the isolated compound is indeed the corresponding acid **74**. We added this discussion to the main manuscript on page 7, lines 8ff.

3

1

To emphasize the usefulness of this enzymatic reaction toward biomedical applications, the author should consider demonstrate how to apply this reaction in making any of the listed compounds (at least one or two examples), e.g. 26, 27, 50 and 64.

Reply: We fully agree with the reviewer that demonstration of application potential is highly important. As we are particularly interested in the chemo-enzymatic synthesis of biomedically interesting natural products, we had already prepared the plant metabolites deoxyvasicinone (**64**) and mackinazoline (**72**) in the initial manuscript. We have now further extended this work by adding a chemo-enzymatic synthesis of antifungal isovasicinone (**66**). The developed synthetic route uses commercially available L-4-Hydroxyproline (**67**), which *via* protected building block **68** can be readily transformed into the desired AsqJ precursor **69** by the synthetic methodology established within the current paper. Transformation of **69** by AsqJ indeed smoothly delivered the desired antifungal **66**, hence demonstrating that both, the substrate synthesis procedures and the discovered AsqJ reactivities, can directly be applied to the production of compounds of biomedical value. We added the respective discussion to the main manuscript, page 6, lines 31 ff. and Figure 8d; the experimental data to the ESI (2.1, 2.2: substrate synthesis of **69**; 3.2, 3.3: conversion of **69** and isolation of **66**).

3

2

Possible mechanisms regarding to quinazolindione was proposed (Figure S45), while hydroxylation was briefly discussed. Because of the majority of enzymes in this super family are classified as hydroxylases. Is hydroxylation associated with this reactivity or is it a result of quenching of cation? While a brief discussion was provided, the author should consider designing experiments to distinguish these possibilities. Notably, the chirality of the hydroxylation seems not determined. The authors should also consider determine the stereochemistry.

Reply: We thank the reviewer for pointing this out to us! We now have conducted in-depth analysis of the stereochemical outcome of the hydroxylation reaction [please also see reply to comment 3 of reviewer 2 above: *We now added these exact experiments as suggested and investigated the corresponding AsqJ product 50 (from 49n) by HPLC on a chiral phase (see ESI, chapter 3.3, Figure S44; manuscript, page 5, lines 43ff.). With this analysis, we observed a ~1:1-mixture of enantiomers of 50, which might hint at a non-enzymatic quenching mechanism. However, it also has to be taken into account that 3-hydroxylated diazepam, such as lorazepam (51), appear to have a very short epimerization half-life time, which was shown to be about 21 minutes for 51.*¹ While we used freshly prepared **50** and conducted downstream chiral analysis very quickly, it cannot be excluded that stereochemical information of an initially stereoselective process is lost before being determinable.] Given the lack of stereochemistry as observed in these analyses, quenching of an intermediate radical/cation by non-enzymatic water addition is a likely explanation.

3

3

The authors should provide kinetic parameters of these reactions. Based upon the HPLC traces the current description, it is difficult for readers to capture the reaction preference. At least, TON (the turnover number) and TOF (the turnover frequency) of some key substrates should be provided.

Reply: Thank you very much for this important suggestion. Prior to submission of the manuscript, we had indeed invested significant amounts of time (i.e., an entire Master's thesis) in the evaluation of methods to determine enzyme kinetic parameters for the different reaction trajectories. As different product profiles are formed by AsqJ depending on the employed substrates – as we describe in the current study – with each of the substrates and product classes having different UV properties, direct measurement of kinetic parameters by a general method monitoring substrate consumption or product formation is not feasible. We therefore attempted to adopt photometric assays that rather monitor alpha-ketoglutarate consumption or succinate production, such as a method reported by Hausinger et al.² This assay is a NADH-coupled photometric assay involving an enzyme cascade allowing indirect detection of succinate. However, in our hands the obtained results never were reliable and reproducible. Even tests trying to detect defined amounts of succinate in the enzyme solution without substrate failed. Therefore, the determination of kinetic parameters of AsqJ was not reliably possible until now and we hence preferred not to report these.

Given the important comment by this reviewer that comparison of relative reaction preferences would be an important addition to the manuscript, we now developed HPLC-based assays for time-dependent consumption measurements for four key substrates, each representing one of the four different major reaction trajectories. This was achieved by measurement of individual HPLC calibration curves for each of these substrates, ultimately allowing accurate consumption measurements by peak area-based calculation for each reaction pathway. These experiments now for the first time provide insights of AsqJ catalytic efficiency for the individual pathways, allowing to assess relative substrate preferences. Our data reveals that phenylalanine-derived substrate **10c** (quinolone pathway) was converted very fast and with very high turnover of >99%. For substrates **9c** (quinazolin(dione) pathway) and **49n** (hydroxylation), the enzyme showed slower transformation, but still high catalytic efficiency (>90% after 6 hours). For proline-derived **54**, the transformation was generally slower, but still reached 46% substrate consumption after 6 hours. We added the new data in the main manuscript as a new own subchapter at the end of results and discussion (main manuscript, page 8, lines 72ff.). The detailed experimental data are shown in the ESI in chapter 3.4.

3

4

The authors explained the observed rearrangement trend, e.g. **30** to **28**, using electronic effect. This is an interesting observation, but should be investigated with caution. The authors also mentioned that this rearrangement can be accelerated under different conditions, e.g. BF₃. To better correlate the observed outcome, it might be helpful to try rearrangement under the reported condition using the available epoxides, e.g. **30**. In addition to acidity, other factors, e.g. solvent, temperature and etc, might influence this reaction.

Reply: Thank you very much for this great suggestion. To investigate the influence of a range of reaction conditions on the fragmentation reaction from epoxides **30** to quinolones **28**, we performed large-scale transformation of substrate **10d** by AsqJ. We then screened 16 different defined fragmentation conditions with systematic variation of the organic solvent (Acetone vs. CHCl₃ vs. MeOH vs. THF), the reaction temperature (r.t. vs. 50 °C), and the acidic reagent (TFA vs. BF₃•OEt₂). While most of these employed conditions led to only minute amounts of the desired quinolone **28d**, with epoxide hydrolysis to the corresponding diol **31d** as the main side reaction, this screening indeed led to the identification of excellent conditions for a clean conversion to quinolones **28**, i.e. by treatment with BF₃•OEt₂ in CHCl₃ at room temperature. Therefore, these additional experiments provided significant additional value to our manuscript – we thank the reviewer for prompting us to conduct these experiments! We added the results of this screening to the main manuscript (page 3, lines 66ff.); experimental details and chromatograms are listed in the ESI in chapter 3.3.

1. Schoetz G., Trapp O. & Schurig V. Dynamic Micellar Electrokinetic Chromatography. Determination of the Enantiomerization Barriers of Oxazepam, Temazepam, and Lorazepam. *Anal. Chem.* **72**, 2758–2764 (2000).
2. Luo L., et al. An assay for Fe(II)/2-oxoglutarate-dependent dioxygenases by enzyme-coupled detection of succinate formation. *Anal. Biochem.* **353**, 69–74 (2006).

REVIEWERS' COMMENTS

Reviewer #2 (Remarks to the Author):

April 3rd, 2023

Dr. Majda Bratovic

Associate Editor in Nature Publications

Nature communications

Dear Dr. Majda Bratovic,

The manuscript (Manuscript ID: NCOMMS-22-50046A) by Tobias A. M. Gulder, et al. entitled "Discovery of extended product structural space of the fungal dioxygenase AsqJ" explores the expanded substrate promiscuity of the fungal dioxygenase AsqJ towards a broad variety of substrates.

This reviewer was convinced the changes in the revised manuscript.

The paper is suitable and recommended for publication in Nature Communications as an Article in the current form.

Reviewer #3 (Remarks to the Author):

I appreciate the authors seriously consider my comments. In my view, the revised manuscript reaches the standard of Nature Communication. If I have to be very critical, one suggesting experiment is to try 18O-water experiment. If formation of 50 is a non-enzymatic process or epimerization, the majority of 50 should contain 18O.

Tabular overview on changes made based on the valuable comments of the reviewers (round 2)

Reviewer #	Comment #	Reviewer Comment
2	1	The manuscript (Manuscript ID: NCOMMS-22-50046A) by Tobias A. M. Gulder, et al. entitled "Discovery of extended product structural space of the fungal dioxygenase AsqJ" explores the expanded substrate promiscuity of the fungal dioxygenase AsqJ towards a broad variety of substrates. This reviewer was convinced the changes in the revised manuscript. The paper is suitable and recommended for publication in Nature Communications as an Article in the current form.
Reply: We thank the reviewer again for his valuable comments to improve the manuscript.		
3	1	I appreciate the authors seriously consider my comments. In my view, the revised manuscript reaches the standard of Nature Communication. If I have to be very critical, one suggesting experiment is to try ^{18}O -water experiment. If formation of 50 is a non-enzymatic process or epimerization, the majority of 50 should contain ^{18}O .
Reply: We thank the reviewer for this suggestion. We performed an additional assay with substrate 49n and H_2^{18}O as main reaction solvent (v/v). By HRMS analysis, we observed a large amount of ^{18}O -labelled product 50-^{18}O (approx. 36% with regards to the corresponding mass counts), which thus indicates that water quenching of a reactive intermediate (11) is the main pathway of formation of 50 . We included this result in the manuscript (page 6, lines 2–9) and details in the ESI (Fig. S45).